# Characterization of the Menin-MLL Interaction as Therapeutic Cancer Target

**DOI:** 10.3390/cancers12010201

**Published:** 2020-01-14

**Authors:** Krzysztof Brzezinka, Ekaterina Nevedomskaya, Ralf Lesche, Andrea Haegebarth, Antonius ter Laak, Amaury E. Fernández-Montalván, Uwe Eberspaecher, Nicolas D. Werbeck, Ursula Moenning, Stephan Siegel, Bernard Haendler, Ashley L. Eheim, Carlo Stresemann

**Affiliations:** 1Bayer AG, Innovation Campus Berlin, Research & Development, Pharmaceuticals, Muellerstrasse 178, D-13353 Berlin, Germany; kbrzezinka1988@gmail.com (K.B.); Ralf.Lesche@bayer.com (R.L.); amaury.fernandez@servier.com (A.E.F.-M.); Uwe.Eberspaecher@bayer.com (U.E.); nicolas.werbeck@bayer.com (N.D.W.); Ursula.Moenning@bayer.com (U.M.); 2Bayer AG, Research & Development, Pharmaceuticals, Muellerstrasse 178, D-13353 Berlin, Germany; Ekaterina.Nevedomskaya@bayer.com (E.N.); Andrea.Haegebarth@bayer.com (A.H.); antoniuster.laak@bayer.com (A.t.L.); Stephan.Siegel@bayer.com (S.S.); Bernard.Haendler@bayer.com (B.H.); Ashley.Eheim@bayer.com (A.L.E.)

**Keywords:** chemical probe, menin-MLL, target validation, cancer, solid tumor

## Abstract

Inhibiting the interaction of menin with the histone methyltransferase MLL1 (KMT2A) has recently emerged as a novel therapeutic strategy. Beneficial therapeutic effects have been postulated in leukemia, prostate, breast, liver and in synovial sarcoma models. In those indications, MLL1 recruitment by menin was described to critically regulate the expression of disease associated genes. However, most findings so far rely on single study reports. Here we independently evaluated the pathogenic functions of the menin-MLL interaction in a large set of different cancer models with a potent and selective probe inhibitor BAY-155. We characterized the inhibition of the menin-MLL interaction for anti-proliferation, gene transcription effects, and for efficacy in several in vivo xenografted tumor models. We found a specific therapeutic activity of BAY-155 primarily in AML/ALL models. In solid tumors, we observed anti-proliferative effects of BAY-155 in a surprisingly limited fraction of cell line models. These findings were further validated in vivo. Overall, our study using a novel, highly selective and potent inhibitor, shows that the menin-MLL interaction is not essential for the survival of most solid cancer models. We can confirm that disrupting the menin-MLL complex has a selective therapeutic benefit in MLL-fused leukemia. In solid cancers, effects are restricted to single models and more limited than previously claimed.

## 1. Introduction

Menin is a scaffold protein encoded by the multiple endocrine neoplasia type 1 (MEN1) gene. Mutations of menin cause a hereditary autosomal dominant tumor syndrome (MEN1 syndrome) leading to tumorigenesis in multiple endocrine organs [1,2]. In humans, menin shows no homology to other known proteins and its precise function is still poorly understood. Significant efforts have therefore been made to identify interacting proteins which might allow for a better understanding of the physiological and tumor-specific functions of menin [2,3,4]. In the nucleus, menin associates with different proteins that activate or repress the transcription of genes involved in multiple cellular processes. Interaction of menin with the JUN family transcription factor JunD has been demonstrated to inhibit transcription of specific target genes. Several MEN1 missense mutations disrupting menin–JunD interaction were identified, suggesting that the tumor-suppressive function of menin in the MEN1 syndrome is connected with the regulation of JunD-target genes [5]. Menin can alternatively interact through the same binding pocket with mixed-lineage leukemia 1 (MLL1/KMT2A/MLL), a histone H3 lysine 4 methyltransferase [4,5]. MLL1 functions as a general transcriptional activator for different gene regulation pathways [6]. In MLL-fused leukemia, which is characterized by different chromosomal translocations at 11q23, menin is essential for maintenance of MLL-associated myeloid transformation. The aberrant gene expression program mediated by the MLL-fusion is abrogated when menin function is reduced in genetic knock-down experiments [7]. Furthermore, other studies recently described important roles of the interaction of menin with MLL in breast cancer, prostate cancer, synovial sarcoma and hepatocellular carcinoma [8,9,10,11]. In breast cancer, a direct interaction of menin with the estrogen receptor (ER) in a hormone-dependent way was described. This interaction promotes MLL recruitment, increased H3K4 methylation and elevates the expression of estrogen target genes [8,12,13]. Furthermore, genetic silencing or chemical inhibition of menin has been shown to reduce proliferation of ER+ breast cancer cell lines via down-regulation of a cancer-specific gene expression program including the estrogen receptor gene (ESR1) itself [8]. In prostate cancer, menin has been identified as an important co-factor for androgen receptor (AR) signaling by direct interaction with the N-terminal domain of the AR and recruitment of the MLL histone methyltransferase activity [9]. Inhibition of menin-MLL interaction with a first generation small-molecule inhibitor showed impaired AR signaling and inhibited the growth of castration-resistant tumors in xenograft experiments in mice [9].

Moreover, menin was linked in miscellaneous studies to different cancer-associated pathways due to its interaction with Akt1, FOXO1, c-Myc or vimentin [14,15,16,17]. These specific interactions result in the regulation of several cancer-relevant phenotypes, so that the interest to explore menin as a potential therapeutic target has recently risen. In this context, it was discussed that interference with the protein-protein interaction surface might be a valid approach to develop small-molecule menin inhibitors [18]. In recent years, several reversible and irreversible small-molecule menin inhibitors (such as MI-503, M-525, MI-136 or MI-2) were reported and used to explore the role of menin in different cancer indications [9,10,13,19,20,21,22,23,24,25]. Nevertheless, an improved chemical probe inhibitor is highly desirable to confirm these findings and further explore the role of menin as a potential cancer target.

Here, we characterize BAY-155 as a novel, potent and selective menin-MLL tool inhibitor [26,27]. We profiled BAY-155 in a large cell line panel for its anti-proliferative activity. Furthermore, we analyzed global gene expression effects in different cancer cells to explore transcriptional effects. Finally, we evaluated the translation to in vivo efficacy in several xenograft models. Overall, we confirmed in our study the important role of menin in MLL-fusion driven leukemia. On the other hand, the activities of BAY-155 in other solid cancer models did not translate into significant anti-proliferative effects. Overall, we conclude that the impact of disrupting the menin-MLL interaction might be restricted to less cancer types than previously reported.

## 2. Results

### 2.1. Characterization of Menin-MLL Tool Inhibitor BAY-155

In search for an optimal chemical probe, we aimed for novel menin inhibitors with potencies and DMPK properties superior to MI-503, which is often used in other studies as the menin-MLL tool inhibitor. Our first approach was to substitute the amino-piperidine moiety of the linker region of MI-503 with suitable spirocyclic amines to obtain optimal binding to the menin protein-protein interaction surface. Supported by docking studies (Figure 1A), we were able to identify several spirocyclic menin inhibitors with increased potency [27,28]. Among these, BAY-155 appeared as the most promising menin inhibitor compared to several other potent spirocyclic amines identified during our optimisation studies.

To better assess the properties of BAY-155, we compared it head-to-head with MI-503 (Figure 1B). First, we compared the compounds ability to interfere with menin-MLL binding in a TR-FRET assay and observed BAY-155 to have an inhibitory IC_50_ of 8 nM, which was 10-fold better compared to that of MI-503. BAY-155 and MI-503 binding to menin was further characterized by an orthogonal biophysical binding assay using Isothermal Titration Calorimetry (ITC) experiments. These experiments confirmed a direct 1:1 binding of BAY-155 to menin with a stoichiometry factor of N = 1.06 and a nanomolar binding affinity with a K_D_ of 75 nM. The MI-503 compound showed comparable binding properties with a K_D_ of 94 nM (Appendix A). Higher potency of BAY-155 translated into improved proliferation inhibition of the MV4;11 and MOLM-13 MLL-rearranged AML cell lines by 2.8 and 6.3-fold, respectively. Interestingly, when testing proliferation inhibition in the non-rearranged AML and B-Cell ALL models HL60 and Jurkat, BAY-155 showed a 13- to 16-fold increased IC_50_ in comparison to MI-503. As for DMPK properties, BAY-155 showed high metabolic stability in rat hepatocytes (CL_blood_ = 1.1 L/h/kg), which translated to an improved maximal bioavailability of 77% in vitro and 15.3% in vivo in comparison to MI-503 (48% and 2.7%). BAY-155 also had a significantly improved Caco2 permeability (26.4 nm/s vs. 0 nm/s for MI-503), which overall, makes it a suitable tool inhibitor for in vivo applications, which can be applied orally. In contrast to MI-503, BAY-155 showed no inhibition of CYP enzymes.

Next we determined whether the improved potency of BAY-155 translated into enhanced cellular mechanistic effects. Activities of menin-MLL inhibitors have so far been robustly validated in MLL-fused AML/ALL cancer cell line models. We therefore tested the impact of BAY-155 and MI-503 on the expression of the stemness-associated gene *MEIS1* and the differentiation genes *MNDA* and *CD11b* in MLL-rearranged MV4;11 and MOLM-13 cells (Figure 1C,D). Indeed, the increased potency of BAY-155 led to a stronger expression down-regulation of the *MEIS1* gene and up-regulation of *CD11b* and *MNDA* genes, in comparison to the effects of MI-503. Additionally, we assessed the selectivity profile of BAY-155 and MI-503 in a panel of assays covering numerous safety pharmacology-relevant targets including G-protein coupled receptors (GPCRs), ion channels and transporters. We observed that BAY-155 had a significantly enhanced selectivity profile in this assay panel compared to MI-503. Treatment with 10 µM BAY-155 inhibited only 7 of the tested proteins by ≥ 50%. Whereas, MI-503 tested at the same concentration inhibited 28 of them (Figure 1E).

MI-503 treatment has previously been reported to reduce menin protein levels in synovial sarcoma models [11]. We performed similar studies with BAY-155 to identify potential differences between both inhibitors (Appendix A). We observed that both MI-503 and BAY-155 were able to reduce menin protein levels in a time- and concentration-dependent manner in VCaP cells (Appendix A). The effects were slightly stronger for BAY-155 and observed in several cell models (Appendix A). We were able to confirm that these effects were dependent on the proteasomal degradation pathway since treatment with MG-132 fully rescued the menin protein level (Appendix A). Overall, we uncovered several improved properties of BAY-155 in comparison to MI-503, which prompted us to use BAY-155 as a tool inhibitor for a broad and independent exploration of the therapeutic potential of menin-MLL inhibition.

### 2.2. Evaluation of Anti-Proliferation Effects after Menin-MLL Inhibition in a Large Cancer Cell Line Panel 

In order to test for general anti-proliferative effects of menin-MLL inhibition, we determined the effect of BAY-155 on 401 cancer cell lines derived from 28 tissues of origin (Figure 2). For the analysis of the results, we defined effects occurring at an IC_50_ below 5 µM as being significant. The cell line panel screen showed that inhibition of menin-MLL did not lead to significant anti-proliferation effects (IC_50_ for BAY-155 above 10 µM) in the vast majority of tested cell line models. Nevertheless, 26% of blood cell-derived models (n = 73) and 30% of liver-derived models (n = 33) showed reduced proliferation after BAY-155 treatment. Among the blood cancer cell lines, all tested MLL-AF4, MLL-AF9 and MLL-TD leukemia cell lines were sensitive to BAY-155. Furthermore, our data support previous results generated after MEN1 knock-out in the Project Achilles data where multiple myeloma and leukemia cells were found to be the most sensitive (Appendix A) [29]. In addition, a few responding cell lines derived from different cancers were identified. Interestingly, the proliferation of cancer models originating from breast, prostate or bone, which were previously reported to be sensitive to MI-503, was not affected by BAY-155 treatment.

### 2.3. Gene Expression Effects after Menin-MLL Inhibition in Cancer Models beyond AML/ALL

According to previous reports, menin-MLL interaction plays a role in multiple cancers due to epigenetic transcriptional regulation and to an essential co-factor role for a number of important master regulators such as AR or ER [8,9]. Therefore, we investigated the gene expression effects of BAY-155 in 13 cancer cell lines to better understand the possible global impact of menin-MLL inhibition. To this end, we performed RNA-seq studies in colorectal (SW1463, LoVo, DLD-1), prostate (22RV1, LNCaP, VCaP), pancreas (PANC-1, MiaPaCa2), breast (BT-474, ZR-75-1, MCF7), bladder (JMSU-1) and kidney (G401) tissue derived cancer cell lines. Menin-MLL inhibition by BAY-155 treatment resulted in moderate expression changes of 10 to 665 genes (log2FC > 1, FDR < 0.1) in a cell line-dependent manner (Figure 3A). In view of the reported role of MLL1 as a co-activator of gene expression it was surprising to find that inhibition by BAY-155 led predominantly to up-regulation of genes (Figure 3A, red bars). More specifically we observed 665 genes to be differentially expressed in the strongest responding bladder cancer cell line JMSU-1, 346 genes in PANC-1, representing an example for a more intermediate response, and 10 genes in ZR-75-1 (Figure 3A). Due to the reported lack of menin protein expression in ZR75-1 [8], we concluded that most observed gene expression effects caused by BAY-155 were specific events linked to menin-MLL inhibition.

To evaluate a general overarching role of menin in transcriptional regulation, which has been for instance reported for the Hox gene cluster [30], we assessed the overlap between differentially expressed genes in different cell lines (Figure 3B). We observed a significant drop in the number of overlapping up-regulated or down-regulated genes, when 2 cell line models were compared. No overlap was observed when four (Figure 3B left for up-regulated genes) or three (Figure 3B right for down-regulated genes) cell models were compared. Altogether, these data suggest that inhibiting the interaction of menin with MLL1 with BAY-155 leads to a cell line-specific response at the gene expression level. We could not observe any evidence of an overarching, menin-driven expression program, which is in contrast to previous reports for e.g., the Hox gene cluster in mice [30].

To further analyze the impact of menin inhibition in a more tissue-specific context, we evaluated gene expression changes in prostate cancer (LNCaP and VCaP) and breast cancer (MCF7) models. We performed Gene Set Enrichment Analysis (GSEA) after BAY-155 treatment and found a significant reversal of androgen-induced gene expression changes, both in VCaP (Figure 4A) and LNCaP (Figure 4D) cells. Despite the highly significant enrichment of R1881-regulated gene sets, the number of genes significantly regulated by BAY-155 that overlapped with genes regulated by R1881 was small in our experiments (Figure 4B,E). More specifically, 36 genes up-regulated by androgen stimulation were repressed after BAY-155 treatment, and 182 genes down-regulated by androgen stimulation were elevated after menin inhibition in VCaP cells (Figure 4B). Similarly, 24 genes up-regulated and 141 down-regulated genes by androgen stimulation were reversed following menin inhibition in LNCaP cells (Figure 4E).

Furthermore, to better understand these effects, we compared gene expression fold-changes after androgen stimulation to changes after androgen stimulation plus BAY-155 treatment. We observed smaller gene expression changes after BAY-155 treatment compared to R1881 induction (Figure 4C,F). Consistent with this, gene expression effects of several prominent AR target genes (*KLK2*, *FKBP5* and *TMPRSS2*) only showed slight expression changes after BAY-155 treatment (Appendix A). In summary, our data confirm a modulating role of menin in AR-mediated gene regulation in prostate cancer [9]. However, the impact of BAY-155 on the androgen-regulated transcriptome was slight and only partially overlapped with the entirety of the effects of androgen signaling.

We also explored the impact of menin-MLL inhibition on gene expression changes in ER-positive breast cancer by using the MCF7 cell line as model. Gene set enrichment analysis of estradiol (E2) up- and down-regulated gene sets revealed a slight reverse regulation of E2 affected genes by BAY-155 (Figure 4G). However, only 26 of E2 up-regulated and 13 of E2 down-regulated genes were significantly changed by BAY-155 treatment (Figure 4H). Furthermore, gene expression changes after BAY-155 treatment compared to E2 induction (Figure 4I), as well as known ER target genes (*EGR3*, *GREB1* and *PGR*) (Appendix A) showed a slight or no expression change after BAY-155 treatment. Overall, our observations with BAY-155 do not provide a strong evidence for an ER/menin-driven expression program, in contrast to previous studies, which used genetic knock-down of menin or the substantially weaker inhibitor MI-2 [8,13].

### 2.4. In Vivo Validation of Menin Inhibition by BAY-155 

In order to assess a potential anti-tumor activity of menin inhibition, we established and tested mouse xenograft models using MV4;11, MOLM-13, KYSE-150, RKO, MCF7 and VCaP human cancer cells. Once or twice-daily oral administration of BAY-155 resulted in strong to moderate reduction of tumor volume and weight of MV4;11, MOLM-13 and KYSE-150 xenograft models. In contrast, BAY-155 had no significant effects on RKO, MCF7 and VCaP xenografted tumors (Figure 5).

In detail, we observed that treatment with BAY-155 (180 mg/kg 2QD p.o.) resulted in regression in tumor volume (T/C < 0.02) with no evidence of body weight loss and a clear concentration-dependent trend for reduced tumor weights in MV4;11 xenograft model (Figure 5A). We further explored BAY-155 in the MOLM-13 xenograft model, where a dose-dependent inhibition of tumor volume and significant reduction of tumor weight after compound administration was observed (Figure 5B). To verify if our in vitro anti-proliferation effects of BAY-155 (Figure 2) translated into significant in vivo efficacy, we selected KYSE-150 and RKO models for in vivo evaluation. BAY-155 treatment (120 mg/kg QD p.o.) resulted in a slight (T/C 0.67) reduction in tumor volume and a significant decrease in tumor weight in the KYSE-150 xenograft model (Figure 5C). Interestingly, no significant anti-growth effects were detectable after BAY-155 administration in the RKO xenograft model, whereas regorafenib treatment resulted in a pronounced reduction of tumor volume and weight (Figure 5D). Finally, we were interested if the effects on gene expression observed for BAY-155 in MCF7 and VCaP cell lines would translate into efficacy in an in vivo experiment. Notably, previous studies using MI-503 or MI-2 for inhibition of the menin-MLL interaction suggested anti-tumor efficacy in breast and castration-resistant prostate cancer. We determined the effects of menin-MLL inhibition in MCF7 and VCaP xenograft models by using BAY-155. The approved drugs tamoxifen and enzalutamide were used as positive controls for these models. BAY-155 treatment had no effects in MCF7 and VCaP xenograft models, whereas tamoxifen and enzalutamide treatment respectively resulted in reduced tumor volume and tumor weight (Figure 5E,F).

### 2.5. Ex Vivo Validation of Cancer Pathway Specific Effects of Menin Inhibition by BAY-155 

Next, we analyzed the ex vivo effects of BAY-155 in these models for a better mechanistic understanding of their mode of action. First, we assessed whether the anti-tumor efficacy was reflected by differential gene expression in the tumor samples from the MV4;11 (Figure 6A) and MOLM-13 (Figure 6B) xenografts. We found a concentration-dependent and significant down-regulation of the MLL-rearranged target gene *MEIS1*, and up-regulation of the differentiation-associated genes MNDA and CD11b upon BAY-155 treatment (Figure 6A,B), confirming the important role of menin in MLL fusion driven de-differentiation [22,26].

In the ex vivo analysis of the VCaP tumor samples for effects on the AR signaling pathway, we found a significant reduction of *FKBP5* and *TMPRSS2* expression after enzalutamide treatment, but no significant changes were seen in the BAY-155 treated study arm (Figure 6C). Furthermore, we also analyzed the MCF7 tumor samples for differential expression of genes from the ER signaling pathway. Tamoxifen treatment led to a significant reduction of *EGR3*, *GREB1* and *PGR1* expression, but no significant effects were observed in samples treated with BAY-155 (Figure 6D). In summary, our in vivo validation of gene expression changes after menin-MLL inhibition in different models confirmed the role of the menin-MLL interaction in MLL-rearranged leukemia. On the other hand, we could not detect a clear in vivo effect of BAY-155 on nuclear hormone receptor signaling even though there was a slight tendency, which did however not reach significance. Taken together with the previous in vitro results, the effects of menin inhibition seen with our tool inhibitor BAY-155 on AR and ER pathways are minor and not robust enough to translate into a significant anti-tumor effect in our models.

## 3. Discussion

Menin is a scaffold protein with multiple protein interaction partners and plays potentially different roles in cancer development. When mutated, it causes the MEN1 syndrome, a rare hereditary endocrine cancer syndrome [31], indicating an anti-proliferative and tumor suppressive role in endocrine tissues. Conversely, an oncogenic function has been identified for menin in MLL-fused leukemia. Here, menin co-recruits the ‘super elongation complex’ (SEC), a driver of aberrant gene expression, through interaction with MLL [7]. Based on these findings, different menin-MLL inhibitors have been tested in MLL-fused AML/ALL and beneficial therapeutic effects have been reported in several pre-clinical models [4,19,22,26].

However, menin-MLL and other menin protein interactions, have been proposed to play additional important roles in cancer models beyond MLL-fused leukemia [8,9,10,11,32]. All these studies have validated the role of menin by knock-down approaches or with different small molecule probes with limited potency and/or selectivity. Additionally, findings of the role of menin in solid cancer models have not been extended since the initial reports and still await further confirmation. Therefore, we have devoted significant resources in identification and characterization of BAY-155, a novel potent inhibitor targeting menin with improved cellular potency and selectivity, and with oral bioavailability. With the improved properties of BAY-155, we were able to re-address and extend on the proposed cancer roles of menin in a more comprehensive chemical biology based target validation [33,34].

We analyzed BAY-155 and MI-503 in a head-to-head comparison. Interestingly, the cross-comparison showed significant differences in the selectivity profiles of both compounds. MI-503 inhibited ~30% of all tested targets in the selectivity panel in contrast to BAY-155, where selectivity was much improved (in only 7/77 cases an IC_50_ value could be reached). Since selectivity has been tested using a high 10 µM concentration, some of the detected off-target activities are probably not relevant in the cellular context. Nevertheless, it cannot be excluded that previously reported effects of MI-503 seen by using higher concentrations were driven by inhibition of other pathways not related to menin.

We used BAY-155 to evaluate the general role of menin in transcriptional regulation in cancer. Menin was reported to be involved in transcriptional activation and repression complexes [2,35]. The interaction with MLL1 and MLL2 [36] as well with c-MYC [17] links menin to transcriptional activation, whereas the interaction with SUV39H1 [37] and JunD [5] associates it with transcriptional repression. Our RNA-seq analysis of samples treated with BAY-155 revealed cell-specific gene expression changes, with very limited overlap between the tested models, which undermines an overarching role of menin in transcriptional regulation. Surprisingly, in the majority of models analyzed more genes were transcriptionally activated than repressed by BAY-155. This suggests that menin has a pronounced role as a transcriptional repressor possibly via interaction with SUV39H1 [37] or JunD [5]. JunD and MLL1 compete for the same binding pocket in menin and it is therefore highly probable that BAY-155 also inhibits formation of menin-JunD complexes. The binding side of the transcriptionally repressive H3K9 methyltransferase SUV39H1 is different from the MLL1 interaction side in menin, so that the potential effects of BAY-155 on menin-SUV39H1 complexes are likely to be indirect. The finding that there are more up- than down-regulated genes after menin inhibition is particularly interesting, since most reports published to date highlighted a gene activating role of menin [35].

Beyond a possible global role of menin as epigenetic regulator, several studies reported more tumor tissue specific mechanisms [5,9,17,22,38]. However, only a very limited number of cancer cell models were tested so far. Herein, we evaluated the menin tool inhibitor BAY-155 for its proliferation effects in a comprehensive cancer cell line panel. Interestingly, BAY-155 showed most pronounced anti-proliferation effects in blood and liver-derived cancer cell lines, whereas only a very limited number of cell lines of other origin responded. Surprisingly, none of the previously described breast [8], prostate [9] or synovial sarcoma [24] models showed a significant anti-proliferative response, which does not support a particularly important role of menin in these cancers. Among the leukemia-derived models, MLL-fused AML/ALL cell lines were classified as the top responders. This trend observed in the in vitro cell proliferation panel was translated into in vivo inhibition of AML-derived xenograft models after oral BAY-155 treatment. Additionally, ex vivo gene expression analysis confirmed previous observations of differentiation marker induction and down-regulation of MLL fusion target genes from in vitro experiments which we and others previously reported [4,22,26].

In summary, by using BAY-155 we could further substantiate previous observations of the prominent role of menin in the pathogenesis of MLL-fused leukemia [4,22], which brings support to the clinical validation of future improved menin inhibitors.

Several previous studies have proposed menin to be critical important for AR- and ER-mediated hormone receptor signaling [8,9,12,13]. In our gene expression analysis in castration-resistant prostate cancer models, we found a slight but significant inhibition of AR receptor regulated genes by BAY-155, which supports a modulating role of menin in this pathway [9]. In ER-positive breast cancer, it was proposed that the menin-MLL complex co-regulates a proliferation-associated gene expression program including *ESR1*, the gene encoding the ER [8]. Surprisingly, findings related to the role of menin in regulating ER-specific gene expression could not be fully confirmed by BAY-155. A possible explanation for this might come from the usage of different validation methods. In previous studies, data had been generated by shRNA knock-down and using a weaker menin tool inhibitor MI-2 (in-house IC_50_ measured for menin-MLL interaction ~1 µM), a parent compound of MI-503. Additionally, we evaluated MCF7 and VCaP xenografts and compared the effects to tamoxifen and enzalutamide, respectively [39,40,41]. Neither, MCF7 nor VCaP tumors exhibited a reduced tumor volume or weight upon BAY-155 treatment. Moreover, ex vivo gene expression analysis of several estrogen- and androgen-regulated genes showed no effect of BAY-155, whereas, significant down-regulation was detectable after tamoxifen (MCF7) or enzalutamide (VCaP) treatment. Therefore, despite some reports on the importance of menin-MLL function in ER-positive breast cancer and AR-positive prostate cancer, we were not able to detect significant anti-tumor effects in pre-clinical models with BAY-155. We found effects on AR gene regulation for BAY-155, but they did not translate into significant anti-proliferative effects in vitro or in vivo. Considering that since the initial reports no further evidence for an impact on these tumors has been published (even not with next generation menin inhibitors [4,23]) early data should be taken with caution, due to the use of potential inaccurate tools.

In addition, based on our cancer cell line panel, we tested the activity of BAY-155 in several in vivo xenograft experiments, which have not been reported before. Both KYSE-150 and RKO cells were highly sensitive to BAY-155 treatment in our in vitro screen (Figure 2) with an IC_50_ in similar range to that for which effects were observed for the MV4;11 and MOLM-13 models. In our analysis for a potential in vivo translation in xenograft experiments, we observed a significant inhibition of KYSE-150 tumor growth. In contrast to that, the RKO model did not show any response to the BAY-155 treatment. This indicates that the limited models which responded in the in vitro proliferation panel do not all behave similarly in an in vivo setting. Several potential reasons could lead to lack of efficacy in a xenograft experiment. One example for this is the in vitro evaluation of BAY-155 on the RKO cells which was performed in a 2D culture setting, while a xenograft experiment utilized a 3D tumor and thus a higher degree of biological complexity. Therefore, the underlying molecular mechanisms (e.g., biomarkers) which resulted in anti-proliferation effects in those models need to be further explored.

## 4. Materials and Methods

### 4.1. Materials

BAY-155 and MI-503 have been synthesized in house as previously described [22,27]. Cell lines: MV4-11, BT474, ZR75-1, RKO, VCaP, 22Rv1, MCF7, G-401, DLD-1, MIA PaCa-2 and PANC-1 cells were obtained from ATCC. MOLM-13, LNCaP, LoVo, SW1463, and JMSU-1 cells were obtained from the Deutsche Sammlung von Mikroorganismen und Zellkulturen (DSMZ, Braunschweig, Germany). All used cells were cultured in the appropriate media and conditions.

### 4.2. Biochemical Assay

The ability of BAY-155 and MI-503 to interfere the menin-MLL interaction was quantified by a TR-FRET assay using recombinant full length, His-tagged menin [42] and a synthetic biotinylated MLL1 a.a. 4–15 (Biosyntan) [28]. Briefly, 2 nM of menin, 0.1 nM to 20 µM test compounds, 50 nM MLL1 peptide and TR-FRET detection reagents (2 nM anti-6XHis-Tb cryptate and 50 nM Streptavidin XL-665, both from Cisbio) were mixed stepwise and allowed to equilibrate for 0.5–4 h at RT in assay buffer (50 mM Tris/HCl pH 7.5, 50 mM NaCl, 0.1% BSA and 1 mM DTT). TR-FRET measurements were performed by excitation of the mix at 337 nm with emission fluorescence readings at 622 nm (donor) and 665 nm (acceptor) in a PHERAstar microtiter plate reader (BMG Labtech). Raw values (acceptor/donor ratios) were normalized between a full binding vehicle control (0% inhibition) and a 100% inhibition control, where menin was replaced by assay buffer. IC50 values were obtained by fitting the normalized data to a four-parameter logistic equation.

### 4.3. Protein Purification

Cloning of expression vectors for recombinant menin was performed as follows. The cDNAs encoding the protein sequences of full-length human menin (O00255-1, 1-615) with an N-terminally fused TEV cleavable GST tag were optimized for expression in eukaryotic cells and synthesized using the GeneArt technology from Life Technologies (Carlsbad, CA, USA). cDNA also encoded att-site sequences at the 5′ and 3′ ends, allowing subcloning into the different destination vectors using the Gateway Technology. Proteins were expressed in *E. coli* BL21 (DE3) following 0.1 mM IPTG induction at 17 °C overnight. For purification, cell pellets were resuspended in buffer A (50 mM Tris pH 7.5, 300 mM NaCl, 2 mM DTT). Cells were lysed using a high pressure Microfluidics apparatus and the cell debris pelleted by centrifugation. The supernatant was applied to a GST column (GE Healthcare, Chicago, IL, USA), washed with two column volumes of washing buffer A and eluted with Buffer B (Buffer A plus 15 mM reduced glutathione). If necessary, the purification tag was cleaved off with the addition of His-TEV protease (1:20 mass ratio) during overnight incubation in buffer A. His-TEV was removed by IMAC column rebinding and final protein purification of either GST-cleaved or uncleaved menin-1 was performed by size exclusion chromatography using a HiLoad 16/60 Superdex 200 (GE Healthcare) column in buffer C (Buffer A but 50 mM NACl).

### 4.4. Isothermal Titration Calorimetry

A solution of purified menin (1–615 with GST-tag cleaved as described above) was prepared in ITC buffer (50 mM sodium phosphate pH 7.5, 50 mM NaCl, 1 mM beta-mercaptoethanol) using a PD-10 desalting column (Sephadex G-25 M, GE Healthcare) and concentrated to a final concentration of 56 µM using an Amicon Ultra-4 centrifugal filter device (Millipore, Burlington, MA, USA). The compound was dissolved in 100% DMSO to a concentration of 10 mM and subsequently diluted to a final concentration of 5 µM in ITC Buffer. Isothermal Titration Calorimetry (ITC) was performed at 25 °C in a Microcal PEAQ-ITC (Malvern Panalytical). 5 µM compound was titrated with 56 μM menin (1–615) in ITC Buffer using injections of 2 μL within 4 s at 100-s intervals and a stirring speed of 1000 rpm. Reference Power was set to 6 μcal/s. Data were collected on high feedback mode. Analysis was performed with a single binding site model in the Microcal PEAQ-ITC Analysis Software (Malvern Panalytical).

### 4.5. Metabolic Stability in Rat Hepatocytes

Liver cells were distributed at a density of 1.0 × 10^6^ cells/mL and treated with 1 µM BAY-155 or MI-503. During incubation, the hepatocyte suspensions were continuously shaken, and aliquots were taken at 2, 8, 16, 30, 45, and 90 min, fixed with cold methanol and stored at −20 °C. Next, samples were centrifuged and the supernatant was analyzed with an LC-MS/MS. The compound half-life was determined from the concentration–time plot. The intrinsic clearances were calculated from the half-life, together with additional parameters (liver blood flow, quantity of liver cells in vivo and in vitro). The hepatic in vivo blood clearance (CL) and the maximal oral bioavailability (Fmax) were calculated using the following parameters: liver blood flow, 4.2 L/h/kg rat; specific liver weight, 32 g/kg rat body weight; liver cells in vivo, 1.1 × 10^8^ cells/g liver; liver cells in vitro, 0.5 × 10^6^/mL.

### 4.6. Permeability Assay

Caco-2 cells were distributed at a density of 4.5 × 10^4^ cells/mL on 24-well insert plates (0.4 μm pore size) and cultured for 15 days with medium change every second or third day. Prior to the permeation assay, the culture medium was replaced with an FCS-free Hepes-carbonate transport buffer (pH 7.2). BAY-155 and MI-503 were dissolved in DMSO and 2 μM of the inhibitor was applied to the apical or basolateral compartment. Analysis of inhibitor content was performed following precipitation with methanol by LC/MS/MS analysis.

### 4.7. Cell Line Proliferation Panel

The characterization of proliferation effects in 401 cancer cell lines was performed on the OmniScreen™ platform (CrownBio). Briefly, cells were seeded at the appropriate density for a five-day growth period and treated with BAY-155 (10 µM maximal concentration with 3.16-fold serial dilutions with 9 dose levels) or left untreated as control. For determination of viability the CellTiter-Glo^®^ Luminescent Cell Viability Assay (Promega, Madison, WI, USA) was used. For absolute IC_50_ determination, a dose-response curve was fitted using a nonlinear regression model with a sigmoidal dose-response using GraphPad Prism 5.0. The surviving rate (%) = (LumTest-LumMedium control)/(LumNon-treated-LumMedium control) × 100%.

### 4.8. Gene Expression

Total RNA was isolated using RNeasy-Plus Mini kit (Qiagen, Hilden, Germany). RNA (1 μg) was reverse transcribed using SuperScript III First-Strand Synthesis SuperMix (Life Technologies) and the generated cDNA was analyzed with commercial primers (Appendix A) by qRT-PCR using TaqMan Fast Advanced Master Mix (Life Technologies). For the RNA-seq study, ZR-75-1, BT474, PANC-1, MiaPaCa-2, 22RV1, JMSU-1, DLD-1, SW1463, LoVo and G401 cells were treated with 5 µM BAY-155 or DMSO for 48 h prior to RNA extraction. After 48 h of 5 µM BAY-155 or DMSO treatment, MCF7, VCaP and LNCaP cells were stimulated for further 12 h with 0.1 nM of E2 (ß-Estradiol) or 1 nM R1881, respectively. MCF7, VCaP and LNCaP cells were cultured in starvation medium (containing 5 µM BAY-155 or DMSO) 6 h before stimulation. RNA was isolated using RNeasy-Plus Mini kit (Qiagen). The purified RNA was used for library preparation (Illumina, San Diego, CA, USA. TruSeq Stranded mRNA Kit) and the generated libraries were sequenced (Illumina, HiSeq2500 HTv4, SR, dual-indexing, 50 cycles).

### 4.9. Data Analysis and Statistical Methods 

RNA-seq reads were aligned to hg38 using STAR aligner. Protein-coding genes with more than 10 reads in more than three samples were used for the analysis (total samples N = 86; genes N = 13,003). DESeq2 was used to find genes differentially expressed upon treatment by the menin inhibitor in each cell line compared to DMSO or hormonal treatment (for MCF-7, LNCaP and VCaP cell lines). Genes with log2 fold change (logFC) above 1 or below −1 and adjusted p-value below 0.1 were considered significantly regulated. GSEA analysis was run on the pre-ranked list based on logFC in prostate (VCaP and LNCaP) and breast (MCF7) cell lines. Data are available at GEO under accession number GSE136272.

### 4.10. Mouse Studies

Tumor cells were injected subcutaneously into the flanks of 6- to 9-week old animals. Efficacy studies in xenograft models were performed at Bayer or EPO GmbH (Berlin-Buch, Germany) using NMRI nu/nu, SCID, BALB/c Nude female mice or SCID male mice (VCaP tumors). All animal studies were conducted in accordance with the animal welfare laws and were approved by the State Office for Health and Social Affairs Berlin (LAGeSo). Experiments were run under the animal license B3600a0007. Compound treatment started at a predefined tumor size or burden or at a predefined day after cell inoculation. BAY-155 was administered orally (p.o.) at various doses and regorafenib, tamoxifen and enzalutamide (resuspended in PEG400/H_2_O 6/4 or PEG400/EtOH 9/1) were used as control treatment for RKO, MCF7 and VCaP xenografts, respectively. Subcutaneous tumor volumes were determined by caliper measurement.

### 4.11. Ex Vivo Analysis

Tumors were harvested, dissociated and total RNA was extracted using the TissueLyser (Qiagen, Hilden, Germany) together with RNeasy-Plus Mini kit according to the manufacturer protocol.

### 4.12. Availability of Data and Materials

The data generated or analyzed during this study are included in the published article and its supplementary files. Gene expression data is available at GEO (https://www.ncbi.nlm.nih.gov/geo/) under accession number GSE136272.

## 5. Conclusions

Based on our independent studies with a novel chemical probe inhibitor, we substantiated that interference with the menin-MLL interaction leads to a specific therapeutic effect in MLL fused leukemia. Other solid cancer models, previously suggested to be significantly responsive to inhibition, were not dependent on menin for survival. These results clearly underline the necessity to cross-validate described phenotypes with selective chemical probes. In summary, the findings of this study with the novel menin tool inhibitor BAY-155 add important critical validation data to the menin-MLL interaction as a therapeutic target in cancer.

## Figures and Tables

**Figure 1 cancers-12-00201-f001:**
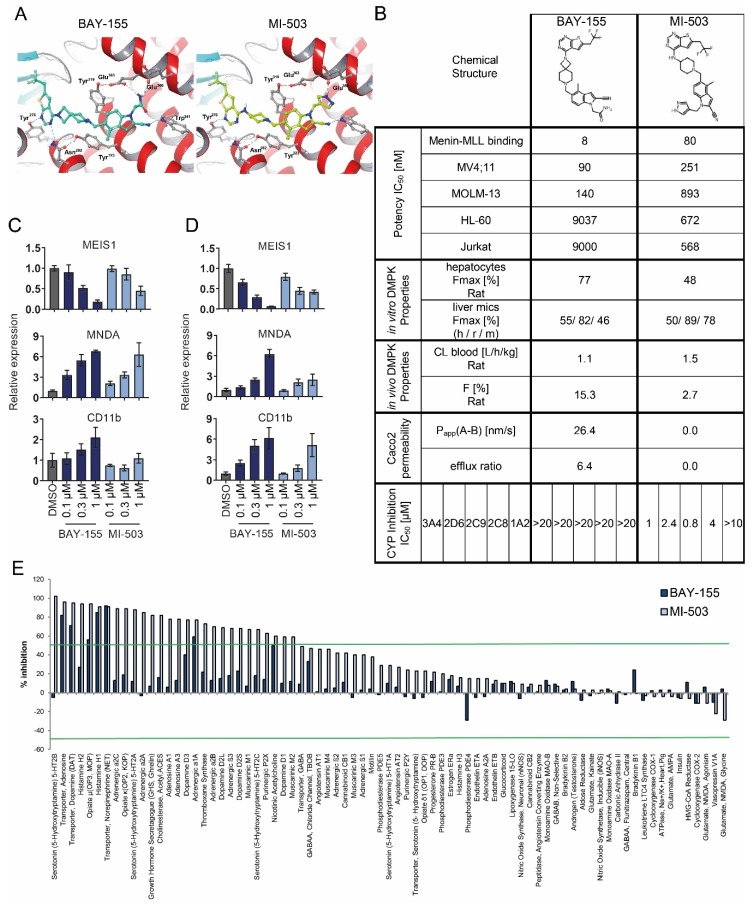
Pharmacological activity and selectivity of the menin-MLL inhibitors BAY-155 and MI-503. (**A**) X-ray structure of menin in complex with compound MI-503 (4X5Y.pdb, **right**) and with docked BAY-155 (**left**). In both binding mode representations, the rotamer of Asn282 has been adapted and taken from 4GQ3.pdb. (**B**) Characterization of drug-like properties of BAY-155 and MI-503. Data presented for BAY-155 have been previously reported [26]. (**C**,**D**) Relative qRT-PCR analysis of genes associated with hematopoietic differentiation in MV4;11 (**C**) or MOLM-13 (**D**) cells after four days of treatment with BAY-155 or MI-503. Data presented are an average of three biological replicates normalized to vehicle control (DMSO). (**E**) Selectivity profile of BAY-155 and MI-503 in a panel of assays covering several safety pharmacology relevant proteins. Inhibitors were tested at a concentration of 10 µM. Green lines indicate 50% of inhibition or activation relative to vehicle control (DMSO).

**Figure 2 cancers-12-00201-f002:**
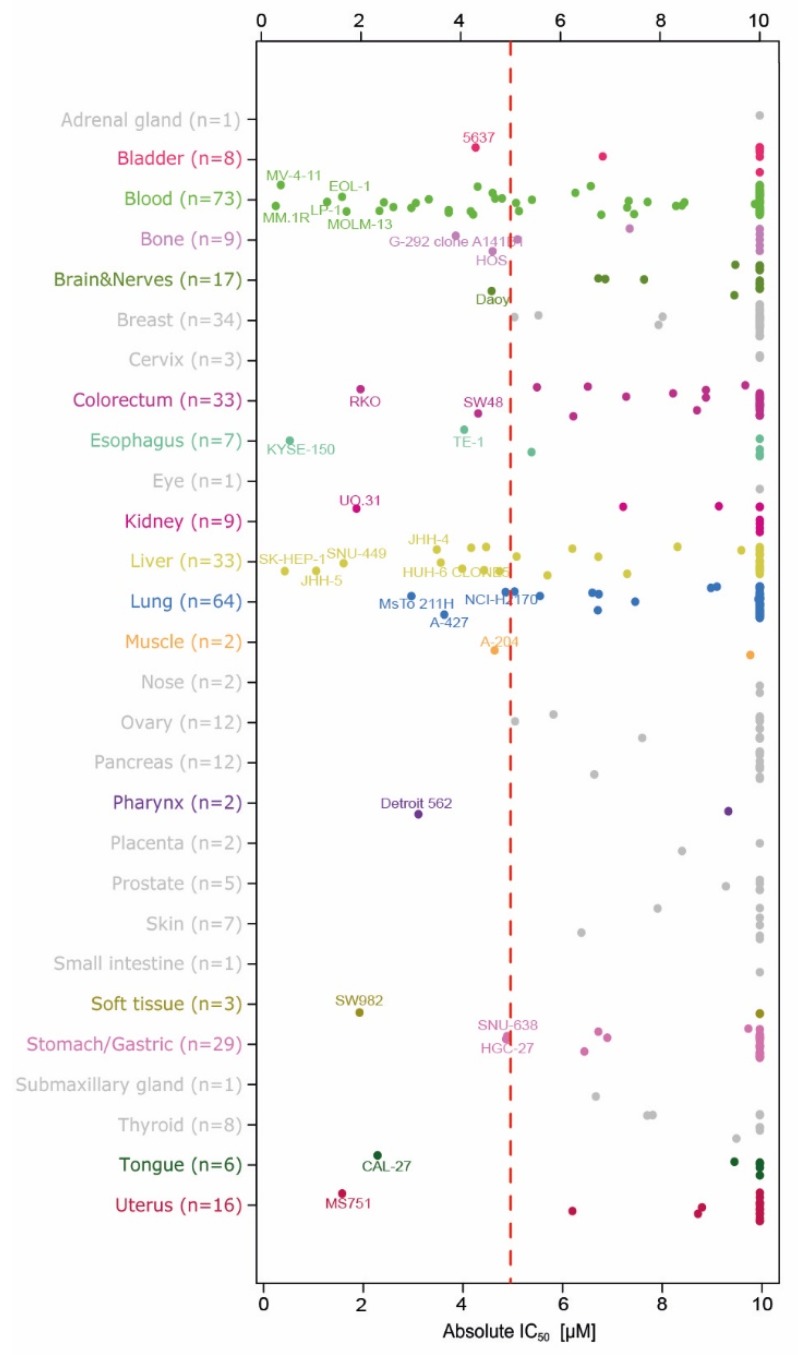
Anti-proliferative activity of BAY-155 in a cancer cell line panel. Sensitive cell lines are indicated to the left of the red dashed line (IC_50_ < 5 µM). For each tissue of origin, up to five highly sensitive cell lines are indicated by name.

**Figure 3 cancers-12-00201-f003:**
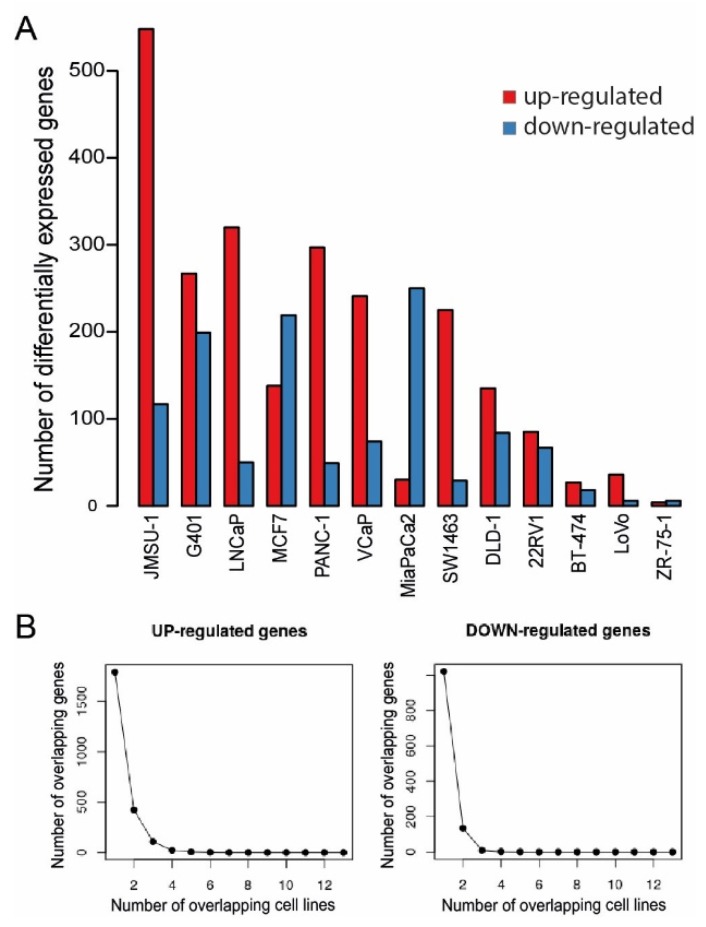
Differential gene expression effects of BAY-155 across several solid cancer cell models. (**A**) Number of genes up-regulated (red bars) and down-regulated (blue bars) upon treatment by BAY-155 in the indicated cell models (logFC > 1 or logFC < −1 and FDR < 0.1). (**B**) Number of overlapping up-regulated (**left**) or down-regulated (**right**) genes in cell lines treated with BAY-155.

**Figure 4 cancers-12-00201-f004:**
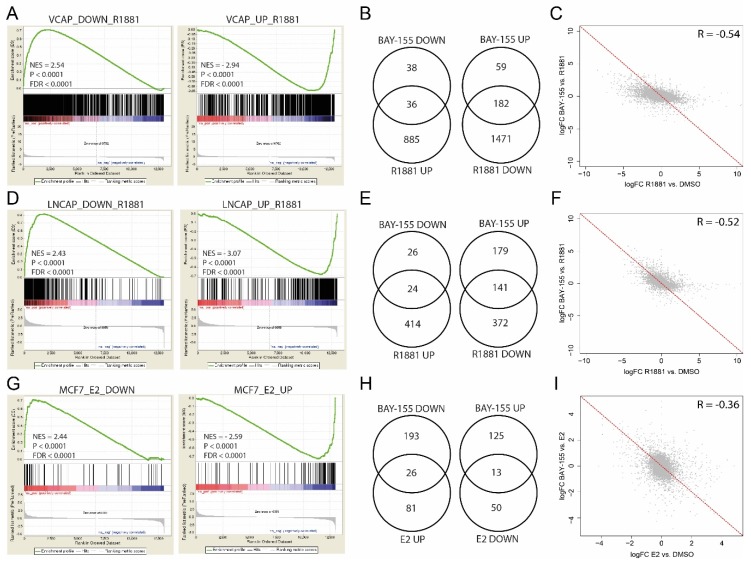
Differential gene expression effects of BAY-155 in prostate and breast cancer models. (**A**,**D**) Gene Set Enrichment Analysis (GSEA) of androgen up-regulated and down-regulated genes after R1881-induction of BAY-155 treated VCaP (**A**) and LNCaP (**D**) cells. (**B**,**E**) Venn-diagrams representing the overlap between BAY-155 and R1881 differentially regulated genes in VCaP (**B**) and LNCaP (**E**) cells. (**C**,**F**) LogFC correlation of R1881-induced gene expression changes and BAY-155 treatment induced gene expression changes in VCaP (**C**) and LNCaP (**F**) cells. (**G**) GSEA of estrogen up-regulated and down-regulated genes after E2-induction of BAY-155 treated MCF7 cells. (**H**) Venn-diagrams representing the overlap between BAY-155 and E2 differentially regulated genes in MCF7 cells. (**I**) LogFC correlation of E2-induced gene expression changes and BAY-155 treatment induced gene expression changes in MCF7 cells. Genes with Log2FC > 1 or < −1 and FDR < 0.1 were considered significantly changed.

**Figure 5 cancers-12-00201-f005:**
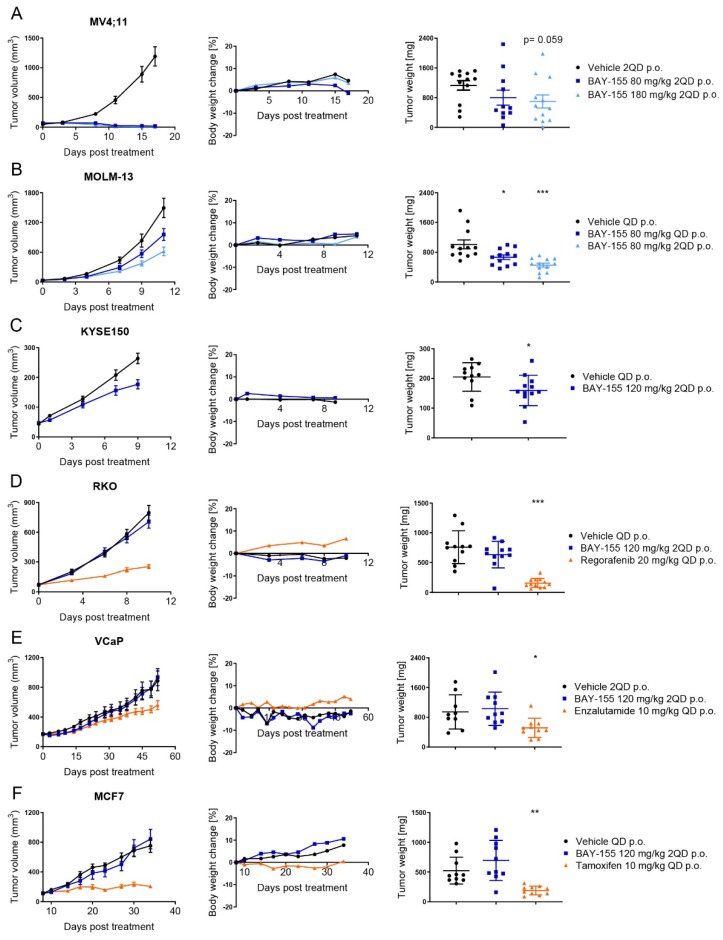
In vivo efficacy of BAY-155 in various tumor models. Tumor volumes (left), body weight change (center) and tumor weights (right) in mice bearing subcutaneous (**A**) MV4;11, (**B**) MOLM-13, (**C**) KYSE150, (**D**) RKO, (**E**) VCaP, (**F**) MCF7 xenografts and treated once or twice daily with the indicated doses of BAY-155. RKO, MCF7 and VCaP xenografts were also treated with regorafenib, tamoxifen and enzalutamide at the indicated doses, respectively. The tumor weight was assessed at the end-point of the study; unpaired *t*-test with Welch’s correction was used for statistical testing. Data are presented as mean value ± SEM, n = 10 or 12 mice per group. Significant difference between vehicle control and treatment group: * *p*-value ≤ 0.05; ** *p*-value ≤ 0.01; *** *p*-value ≤ 0.001.

**Figure 6 cancers-12-00201-f006:**
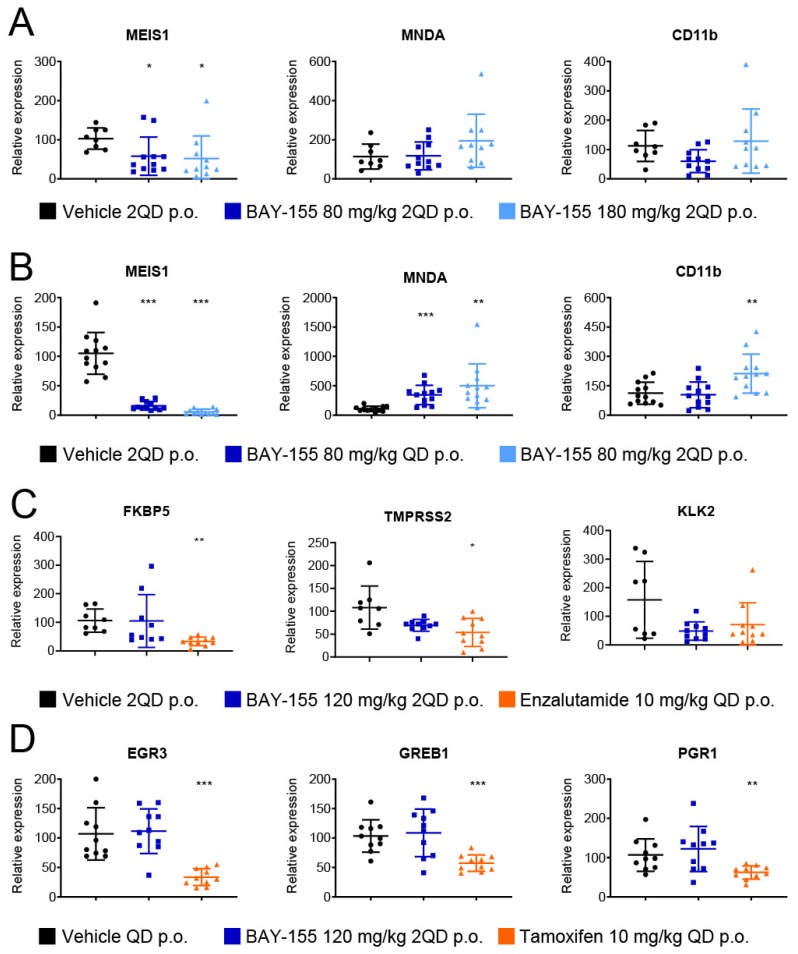
Mechanistic mode of action effects of BAY-155 ex vivo. (**A**,**B**) Relative expression level of *MEIS1*, *MNDA* and *CD11b* after 17 or 11 days of treatment with BAY-155 or vehicle at the indicated dose in MV4;11 (**A**) or MOLM-13 (**B**) xenografts, respectively. (**C**) Relative expression level of *FKBP5*, *TMPRSS2* and *KLK2* after 52 days of treatment with BAY-155, enzalutamide or vehicle at the indicated dose in VCaP xenografts. (**D**) Relative expression level of *EGR3*, *GREB1* and *PGR1* after 27 days of treatment with BAY-155, tamoxifen or vehicle at the indicated dose in MCF7 xenografts. Data were normalized to 18S RNA and the average vehicle control. Significant difference between vehicle control and treatment group: * *p*-value ≤ 0.05, ** *p*-value ≤ 0.01, *** *p*-value ≤ 0.001, two-sided *t*-test.

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
