# Peer review of "Characterization of the Menin-MLL Interaction as Therapeutic Cancer Target"

_cancers, 2020, doi:10.3390/cancers12010201_

Round 1

Reviewer 1 Report

The authors characterize the anti-cancer activity of a menin inhibitor (BAY-155) derived from and superior to the previously published MI-503. They profiled the compound in a panel of 401 cancer cell lines representing 28 human tissues, analyzed its effect on gene expression in 13 cancer cells, conducted mechanistic studies in prostate and breast cancer cells, and tested its in vivo efficacy in several xenograft models. While they do confirm that menin is a promising target in MLL-fusion driven leukaemia, they see little effect on solid tumours except liver, which is in contrast with previous high impact reports in the literature.  As such, this work is an important word of caution regarding the validity of menin as a cancer target outside MLL-fusion leukaemia.

This manuscript describes a high-quality chemical probe against menin which is superior to previously reported compounds, and confirm previous claims while questioning others regarding this potentially important therapeutic target. Results are clear and convincing, the manuscript is well written, and deserves publication in Cancers, following the minor revisions detailed below.

Line 16: Hoverer -> However

Line 64: There were several generations of menin inhibitors published over many years in the MI-xxx series. “first generation” should therefore be deleted  from this sentence.

Line 66: The statement regarding the limitations in potency and selectivity of past inhibitors was not substantiated before this work, and does not belong in the introduction

Figure 1E: The biochemical selectivity profile is conducted at 10 µM, which is not a concentration at which any of these proteins will see the compound in cell assays. The profile nevertheless indicates a lower propensity for BAY-155 to hit off-targets. This should be clarified in the text, in the  lines 310-315 section.

Line 113 e -> (E)

Line 333: the authors state that they found no significant anti-proliferative response in liver cancer models. This seems to be in contradiction with Figure 2, and should be clarified.

Author Response

Dear reviewer,

Thank you very much for your positive feedback and input to further improve the quality of the publication.

You will find below our point-by point responses to your comments:

“Line 16: Hoverer -> However”

We changed it accordingly.

“Line 64: There were several generations of menin inhibitors published over many years in the MI-xxx series. “first generation” should therefore be deleted from this sentence.”

We agree and changed it accordingly.

“Line 66: The statement regarding the limitations in potency and selectivity of past inhibitors was not substantiated before this work, and does not belong in the introduction”

That´s true. We deleted that part.

Figure 1E: The biochemical selectivity profile is conducted at 10 µM, which is not a concentration at which any of these proteins will see the compound in cell assays. The profile nevertheless indicates a lower propensity for BAY-155 to hit off-targets. This should be clarified in the text, in the lines 310-315 section.

Especially taking factors like protein binding into account. Nevertheless, testing at this high concentration gives a good impression on the underlying selectivity profile of the compound. Most of those (weaker) activities might not be relevant in the cellular context. But having a broad activity in 30% of all tested targets might indicate for additional off-target activities (not tested in the panel) at potentially lower concentrations. MI-503 has been used before in cellular assays in the 5 µM range (e.g. Malik et al. 2015), which might be a relevant concentration for hitting additional off-targets. We changed the paragraph and included that most selectivity findings might not be relevant for the cellular assays. We still think that the different selectivity profile explains differences between MI-503 and BAY-155.

“Line 113 e -> (E)”

We changed it accordingly.

“Line 333: the authors state that they found no significant anti-proliferative response in liver cancer models. This seems to be in contradiction with Figure 2, and should be clarified.”

Thank you for finding this contradiction. In addition, the citation in this sentence for liver was wrong (should be [10] Kempinska et al., Molecular cancer therapeutics 2018, 17, 26-38.) Originally, we included also liver in this statement here since the models described in Kempinska et al. (e.g. HepG2) did not respond in our study on the proliferation level after treatment with BAY-155. But indeed, as mentioned in the sentence before, several liver models were inhibited on the proliferation level by BAY-155 with single digit µM IC50. Further studies will have to be done to clarify underlying mechanisms. Therefore, we deleted liver in line 333, since this might still be a potential indication for menin inhibitors.  

Reviewer 2 Report

The manuscript provides a comprehensive screen of a potential menin-MLL interaction inhibitor in various cancer models. It delivers important and new information on which cancer types are responsive to this type of inhibition and reevaluates previous findings in the literature. It is well written and the results are clearly explained, the figures are accurate and mostly adequate. I have only some minor comments.

1. It is not entirely clear what was the rationalization behind chosing specific cell lines for RNA-seq studies. Based on the results presented on Figure 2., many of the cell lines tested were not among the ones responsive to BAY-155 and no blood cancer cell line was included in this study.

2. The font size in some cases is very small, making some figures difficult to read. E.g. Labeling of residues in the crystal structures on Figure 1., names of the sensitive cell lines on Figure 2., labels on Figure 3.

3. The sentence: "No overlap was observed when four (Figure 3B left) or three (Figure 3B right) cell models were compared." (lines 181-183) is a little confusing, since according to the legend, the left panel of Figure B represents the up-regulated genes and the right panel represents the down-regulated genes.

4. There is a mistake in line 248, where Figure 3. is mentioned, while it is Figure 2. that represents the anti-proliferation effects of BAY-155.

5. While discussing the results, it is mentioned that menin might have a pronounced role as a transcriptional repressor, suggesting SUC39H1 and JunD as possible mediators. Reflecting to this suggestion, it would be beneficial for the manuscript if the authors dicussed the known details of menin-SUV39H1/JunD interactions. Mainly, how the interaction surfaces are related to the menin-MLL interaction surface and whether BAY-155 has similar inhibitory effects on those interactions as it has on menin-MLL1 binding.

Author Response

Dear reviewer,

Thank you very much for your positive feedback and input to further improve the quality of the publication.

You will find below our point-by point responses to your comments:

It is not entirely clear what was the rationalization behind chosing specific cell lines for RNA-seq studies. Based on the results presented on Figure 2., many of the cell lines tested were not among the ones responsive to BAY-155 and no blood cancer cell line was included in this study. The rationale for choosing models for the RNA-seq study was not focused on responsive cell lines on the proliferation level. We included several different cancer models derived from different tissues to answer the question, if there is a general expression pattern which is commonly regulated by the menin-MLL interaction. Focusing only on models which are also responding on the proliferation level would potentially bias the results towards proliferation associated gene expression pathways. Therefore, we included breast and prostate cancer models which have been characterized before (MCF-7, VCaP, 22RV1, LNCaP, ZR75-1) to validate those results, with additional cell lines derived from different other cancer tissues origins (colorectal, pancreas, bladder, rhabdoid). Blood cancer cell lines have been characterized extensively before by others and us (see e.g. [26]) and some findings (e.g. MLL-fusion target gene/differentiation gene regulation) was reproduced independently in this study in Figure 1C. The font size in some cases is very small, making some figures difficult to read. E.g. Labeling of residues in the crystal structures on Figure 1., names of the sensitive cell lines on Figure 2., labels on Figure 3. Font sizes have been increased in the mentioned Figures. The sentence: "No overlap was observed when four (Figure 3B left) or three (Figure 3B right) cell models were compared." (lines 181-183) is a little confusing, since according to the legend, the left panel of Figure B represents the up-regulated genes and the right panel represents the down-regulated genes. The content here is correct. Here the perspective is important. We focus in the analysis only on up-regulated genes or down-regulated genes. By comparing 4 cell lines to each other you cannot find any overlaps in the up-regulated genes anymore (left panel). Same is true by comparing three cell lines focusing on the down-regulated genes (right panel). We added for clarification again the up-regulation or down-regulation. There is a mistake in line 248, where Figure 3. is mentioned, while it is Figure 2. that represents the anti-proliferation effects of BAY-155. The mistake was fixed. While discussing the results, it is mentioned that menin might have a pronounced role as a transcriptional repressor, suggesting SUC39H1 and JunD as possible mediators. Reflecting to this suggestion, it would be beneficial for the manuscript if the authors dicussed the known details of menin-SUV39H1/JunD interactions. Mainly, how the interaction surfaces are related to the menin-MLL interaction surface and whether BAY-155 has similar inhibitory effects on those interactions as it has on menin-MLL1 binding. We extended the discussion part of the JUND and SUV39H1 interaction in the manuscript accordingly. JUND interacts with menin through the same binding pocket as does MLL1 (and also MLL2) (see [5] Huang et al. 2012). Menin represses JUND-mediated transcriptional activitiy via inhibition of JUND phosphorylation through sequestering JUND from JNK. MLL1 and JUND1 are competing for the same menin binding pocket. Inhibitors binding to the menin binding site (like BAY-155) therefore are inhibiting binding to MLL1 and JUND with potentially different transcriptional outcomes. Competing JUND away from menin might trigger phosphorylation of JUND and activation of JUND regulated genes (see also Agarwal,S.K. et al., Menin interacts with the AP1 transcription factor JunD and represses JunD-activated transcription. Cell 96, 143–152 1999). Direct binding of menin to the H3K9 specific methyltransferase SUV39H1 has been shown by CO-IP experiments and binding side was mapped to be distinct from the MLL1 binding side (see [37]). Therefore, inhibitors which address the MLL1/JUND binding side might not directly affect binding to SUV39H1. Nevertheless, changing formation of menin-MLL1/MLL2/JUND complexes by blocking this binding side might also impact formation of functional menin-SUV39H1 complexes. But the interaction of menin with SUV39H1 are far less understood and further studies are needed to fully understand this mechanism.